# Isolation of Functional SARS-CoV-2 Antigen-Specific T-Cells with Specific Viral Cytotoxic Activity for Adoptive Therapy of COVID-19

**DOI:** 10.3390/biomedicines10030630

**Published:** 2022-03-09

**Authors:** Estéfani García-Ríos, Alejandra Leivas, Francisco J. Mancebo, Laura Sánchez-Vega, Diego Lanzarot, José María Aguado, Joaquín Martínez-López, María Liz Paciello, Pilar Pérez-Romero

**Affiliations:** 1National Center for Microbiology, Instituto de Salud Carlos III Majadahonda, 28221 Madrid, Spain; egarcia@isciii.es (E.G.-R.); fj.mancebo@isciii.es (F.J.M.); mpperez@isciii.es (P.P.-R.); 2Department of Science, Universidad Internacional de Valencia—VIU, PintorSorolla 21, 46002 Valencia, Spain; 3Department of Hematology, Hospital Universitario 12 de Octubre—Universidad Complutense, Instituto de Investigación Sanitaria Hospital 12 de Octubre (imas12), Avenida Córdoba s/n, 28041 Madrid, Spain; alejandraleial@gmail.com (A.L.); lausanve@hotmail.com (L.S.-V.); jmarti01@med.ucm.es (J.M.-L.); 4H12O-CNIO Haematological Malignancies Clinical Research Unit, Spanish National Cancer Research Centre, 28041 Madrid, Spain; 5MiltenyiBiotech, 28223 Madrid, Spain; diego@miltenyi.com; 6Unit of Infectious Diseases, Hospital Universitario 12 de Octubre, Instituto de Investigación Sanitaria Hospital 12 de Octubre (imas12), Avenida Córdoba s/n, 28041 Madrid, Spain; jaguadog1@gmail.com; 7Department of Medicine, Universidad Complutense, 28041 Madrid, Spain

**Keywords:** SARS-CoV-2, T-lymphocytes, cytotoxicity, M protein

## Abstract

In order to demonstrate the feasibility of preparing clinical-grade SARS-CoV-2-specific T-cells from convalescent donors and the ability of these cells to neutralize the virus in vitro, we used blood collected from two COVID-19 convalescent donors (before and after vaccination) that was stimulated with specific SARS-CoV-2 peptides followed by automated T-cell isolation using the CliniMacs Prodigy medical device. To determine cytotoxic activity, HEK 293T cells were transfected to express the SARS-CoV-2 M protein, mimicking SARS-CoV-2 infection. We were able to quickly and efficiently isolate SARS-CoV-2-specific T lymphocytes from both donors before and after they received the Pfizer-BioNTech vaccine. Althoughbefore vaccination, the final product contained up to 7.42% and 30.19% of IFN-γ+ CD3+ T-cells from donor 1 and donor 2, respectively, we observed an enrichment of the IFN-γ+ CD3+ T-cells after vaccination, reaching 70.47% and 42.59%, respectively. At pre-vaccination, the isolated SARS-CoV-2-specific T-cells exhibited cytotoxic activity that was significantly higher than that of unstimulated controls (donor 2: 15.41%, *p*-value 3.27 × 10^−3^). The cytotoxic activity of the isolated SARS-CoV-2-specific T-cells also significantly increased after vaccination (donor 1: 32.71%, *p*-value 1.44 × 10^−5^; donor 2: 33.38%, *p*-value 3.13 × 10^−6^). In conclusion, we demonstrated that SARS-CoV-2-specific T-cells can quickly and efficiently be stimulated from the blood of convalescent donors using SARS-CoV-2-specific peptides followed by automated isolation. Vaccinated convalescent donors have a higher percentage of SARS-CoV-2-specific T-cells and may be more suitable as donors. Although further studies are needed to assess the clinical utility of the functional isolated SARS-CoV-2-specific T-cells in patients, previous studies using the same stimulation and isolation methods applied to other pathologies support this idea.

## 1. Introduction

The pandemic spread of severe acute respiratory SARS-CoV-2 has caused more than 100 million cases of COVID-19 and 2 million deaths [1,2]. The arrival of effective vaccines has significantly reduced this global health crisis [3], contributing to a decrease in the incidence of severe cases. However, the emergence of new SARS-CoV-2 variants not fully covered by the immune response upon vaccination has recently increased the number of hospital admissions and severe cases of COVID-19.

The specific T-cell response, consisting of both CD4+ and CD8+ T-cells, plays a major role in controlling infection by different viruses [4], such as the measles virus [5], cytomegalovirus (CMV) [6,7,8], hepatitis C virus (HCV) [9], and HIV [10,11]. More recently, the T-cell immune response has been increasingly recognized as a key factor in controlling virus clearance and the severity of COVID-19, in addition to the humoral immune response [12,13,14,15]. In this sense, strong and durable CD4+ and CD8+ T-cell responses [12,16] have been observed in many cases of SARS-CoV-2-infected patients, highlighting their predominant role in SARS-CoV-2 control and dispatch.

Although SARS-CoV is not known to productively infect T-cells, altered antigen-presenting cell function and impaired dendritic cell migration result in the reduced priming of T-cells, likely contributing to fewer virus-specific T-cells in the lungs [17,18,19]. These factors contribute to the severity of the respiratory disease, leading to acute respiratory distress syndrome (ARDS) presenting with respiratory failure that requires mechanical ventilation, shock, multi-organ failure, and death.

Unfortunately, COVID-19 treatment is still an unmet medical need [20,21]. Passive humoral immunity can be immediately acquired by the infusion of plasma from convalescent donors [22]. Furthermore, different studies administrating convalescent plasma infusion to COVID-19patients have shown benefits, preventing progression to non-invasive ventilation or high-flow oxygen, invasive mechanical ventilation or extracorporeal membrane oxygenation, or death at 28 days. However, patients with severe symptoms admitted to an intensive care unit demonstrated no benefits [23].

We hypothesize that lymphocytes from convalescent patients may contain SARS-CoV-2-specific T-cells, which can be purified and used for adoptive therapy using a cell production method developed to purify virus-specific cells. Among the methods used, the CliniMACS Prodigy system facilitates the culture and purification of primary cells, such as CAR-T-cells, previously used to treat hematological malignancies [24,25], or regulatory T-cells to attenuate graft-versus-host disease [26]. Furthermore, the CliniMACS Prodigy system has been widely used to generate specific T-cells upon stimulation with viral antigens from CMV or the Epstein–Barr virus [25,27,28,29]. In addition, in the last decade, few studies have evidenced the stimulation of antigen-specific T-cells upon stimulation with viral-specific peptides [30,31].

The aims of this study were to demonstrate the feasibility of preparing clinical-grade SARS-CoV-2-specific T-cells from convalescent donors and to demonstrate the ability of these cells to neutralize the virus in vitro. Additionally, we intendedto study whether SARS-CoV-2-specific functional T-cells are boosted after vaccination with an mRNA vaccine.

## 2. Materials and Methods

### 2.1. Patient Samples

Samples collected from two donors with confirmed SARS-CoV-2 infection (documented by RT-PCR) were studied before and after receiving the Comirnaty SARS-CoV-2 mRNA Pfizer-BioNTech vaccine. Donor 1 had SARS-CoV-2 infection in April 2020 with radiological-documented pneumonia, while donor 2 had mild symptoms in September 2020. Both donors developed detectable immunoglobulin G (IgG) antibodies after clearing the infection, as documented by a negative RT-PCR result. This study was approved by the local Ethics Committee for Clinical Research and was conducted in accordance with the Declaration of Helsinki and the Guidelines for Good Clinical Practice.

### 2.2. T-Cell Reactivity Testing

A total of 10 mL of whole peripheral blood was collected from each donor in EDTA tubes. To test the reactivity of donor T lymphocytes against the different SARS-CoV-2 peptides (M, N, and S), fresh peripheral blood mononuclear cells were isolated by centrifugation (2400 rpm, 20 min) using a density gradient with Ficoll-Paque^®^ (Cytiva Life Sciences, Wein, Austria). Cells were later washed with PBS (Lonza, Basel, Switzerland) and diluted to 10 × 10^6^ cells/mL with RPMI-1640 medium (Biowest, Nuaillé, France) supplemented with 10% human AB serum (Sigma-Aldrich, Merck, St. Louis, MO, USA), and 100 μL/well was seeded into 96-well microplates. Non-human proteins should not be added to avoid non-specific stimulation. Cells were stimulated for 4 h using the peptivator tool, a pool of lyophilized peptides, consisting mainly of 15-mer sequences with 11 amino acids overlapping and covering the complete protein sequence. This mixture of overlapping peptides was used for each of the S, N, and M proteins (0.6 nmol) of SARS-CoV-2 (MiltenyiBiotec, BergischGladbach, Germany). Non-specific stimulation (Cytostim, MiltenyiBiotec, BergischGladbach, Germany) was used as a positive control. Cytostim is an antibody-based reagent that stimulates effector and memory T-cells by TCR binding and crosslinking them to a major histocompatibility complex of antigen-presenting cells. It acts as an antigen for CD4+ and CD8+ cells, causing their stimulation, which triggers the release of cytokines (IFN-γ, IL-2, IL-4, Il-5, IL-10, TNF-α, and IL-17A) and upregulation of activation markers (CD25 and CD69). Unstimulated cells were used as a negative control. After 2 h, 10 μg/mL of Brefeldin A (Sigma-Aldrich, Merck, St. Louis, MO, USA) was added to stop IFN-γ secretion. Finally, cells were washed and stained with fluorescent-conjugated antibodies.

### 2.3. Flow Cytometry

Cells were analyzed in a FACS Canto II flow cytometer (BD Bioscience, San Jose, CA, USA) by quantification of CD3+ IFN-γ+ cells by flow cytometry. Analysis was based on doublet exclusion (selection by FSC-A and FSC-H) and lymphocyte population selection (by FSC-A and SSC-A selection). A minimum of 100,000 total events were analyzed. Alist of the fluorochrome-labeled antibodies used inthe study is shown in Table 1, and the gating strategy is reported in Figure 1A. Incubation with antibodies labeling the cell surface, namely, CD3 (1/100 dilution), CD4 (1/100 dilution), and CD8 (1/100 dilution) antibodies, was performed for15 min at room temperature in 100 µL of PBS + 3% FBS + 3 mM EDTA. Cells were washed with PBS (2300 rpm, 3 min), and then cells were fixed (with fix solution, 3.7% formaldehyde), permeabilized (Inside Stain Kit, MiltenyiBiotec, BergischGladbach, Germany), and stained with anti-IFN-γ-PE (MiltenyiBiotec, BergischGladbach, Germany) 1/50 dilution in Perm buffer (100 µL) for 20 min. Finally, cells were washed again with Perm buffer (2300 rpm, 3 min). Antigen stimulation was considered positive, as established by the manufacturer’s protocol, when the percentage of CD3+ IFN-γ+ cells was higher than 0.1% and the positive cell count was more than 2-fold compared with the unstimulated control (established threshold of the minimum percentage and cell number detected with the non-specific control). CD4+ and CD8+ T-cell subsets were also analyzed by flow cytometry.

### 2.4. Lymphapheresis

Using a Spectra Optia (Terumo BCT) and the CMNC program, the procedure was performed by peripheral venous puncture and by processing between 1.5 and 2 donor volume until a product with at least 1 × 10^9^ total leukocytes and a 2–3% hematocrit was obtained. The collection flow rate was 1.0 mL /min, and the maximum inflow rate was 75 mL /min. Adenine-citrate-dextrose (ACD) was used as an anticoagulant in a 12:1 ratio (ACD: blood). After apheresis, a differential count using a DXH 520 analyzer was performed in order to determine the total nucleated cells (TNCs) in the apheresis product. The procedure was repeated after the donors were vaccinated using two doses of the Pfizer mRNA vaccine.

### 2.5. SARS-CoV-2 Antigen-Specific T-Cell Production

Briefly, 1 × 10^9^ cells from lymphapheresis were diluted with PBS-EDTA buffer and 0.5% human serum albumin to 50 mL and stimulated using overlapping peptides of SARS-CoV-2 (selected from the reactivity testing experiments), covering the complete sequence domains of the M protein in donor 1, and proteins M, N, and S (immunodominantsequence) in donor 2 (Peptivator pool, MiltenyiBiotec). Peptide pools bind to MHC class I and class II complexes, stimulating both CD4+- and CD8+-specific T-cells (60 nmol/peptide wasused for the stimulation of T-cells). Stimulated T-cells were labeled with the Catchmatrix Reagent (MiltenyiBiotec, BergischGladbach, Germany), which contains bispecific antibodies for CD45 and IFN-γ, secreted upon stimulation. The cell-surface-bound IFN-γ was targeted by the Enrichment Reagent, which contains IFN-γ-specific antibody conjugated to superparamagnetic iron dextran particles (MACS^®^MicroBeads), allowing subsequent immunomagnetic cell separation. The microbead-labeled cells (CCS positive fraction) were retained in the built-in magnetic column and further eluted with saline serum into the target T-cell bag. The entire cell manufacturing process took only 12 h. The obtained cells were analyzed by flow cytometry as described in the previous section to confirm the INF-γ antigen-specific T-cell enrichment.

### 2.6. Cells and Cloning

HEK 293T cells were obtained from the American Type Culture Collection (ATCC; Manassas, VA, USA) and were cultured in flasks with DMEM supplemented with 10% fetal bovine serum (FBS), 20mM glutamine (Lonza), 10 units of penicillin, and 10 µg of streptomycin (Lonza).

The ORF encoding the SARS-CoV-2 membrane (M) protein was amplified by RT-PCR using SARS-CoV-2 RNA as a template, with the SuperScript™ IV One-Step RT-PCR System (Invitrogen; Waltham, MA, USA). The PCR product and the pcDNATM3.1/myc-His (−) (5.5 kb) vector (Invitrogen) were digested with the *Xho*I and *Kpn*I restriction enzymes (FastDigest enzymes, Thermo), ligated (Ligase, ThermoScientific; Waltham, MA, USA), and transformed into the XL10 Gold *E.coli* chemically competenT-cells. The constructed plasmid was verified by sequencing. The oligonucleotides used are listed in Table 2.

### 2.7. Cytotoxicity Assay

We determined the capacity of the SARS-CoV-2-specific T-cells to recognize and lyse HEK 293T cells expressing the SARS-CoV-2 M protein in the cell membrane (T-cell cytotoxic activity). Thus, HEK 293 T-cells were transfected with the plasmid pcDNATM3.1/myc-His containing the M protein gene. As a control, HEK 293 T-cells were transfected with the empty pcDNATM3.1/myc-His plasmid. The day before transfection, 2 × 10^4^ HEK 293Tcells were seeded per well in a 96-well plate and incubated until 80–90% confluency was reached. The next day, cells were transfected using the Lipofectamine 2000 reagent (Invitrogen, Waltham, MA, USA) following the manufacturer’s instructions using 0.2 μg of the pcDNATM3.1-M-gene construction diluted in 25 μL of OptiMEM (ThermoFisher Scientific, Waltham, MA, USA. To calculate the percentage of HEK 293Tcells expressing M protein after transfection, we determined the transfection efficiency using a GFP protein reporter plasmid in parallel.

The next day, the HEK 293Ttransfected cells were co-cultured with 2 × 10^5^ stimulated or unstimulated T-cells from two COVID-19 convalescence donors in 50:50 DMEM:RPMI medium and incubated for 24 h at 37 °C and 5% CO_2_. Cytotoxic activity of donors’ stimulated and unstimulated (used as negative control) T-cells was measured 24 h after co-culture with the transfected HEK 293Tcells using the CyQUANT™ LDH Cytotoxicity Assay (ThermoFisher Scientific, Waltham, MA, USA). At least three replicates of each experiment were performed.

### 2.8. RNA Extraction and RT-PCR

Expression of SARS-CoV-2 M protein was quantified in the HEK 293Ttransfected cells versus the HEK 293Tempty pcDNATM3.1 transfected cells used as a control.After transfection, cells from a well of a 6-well plate were collected and pelleted. RNA was extracted from the transfected cells using the E.Z.N.A.^®^Total RNA Kit I (Omega Biotek; Norcross, GA, USA) according to the manufacturer’s protocol. Then, 1 ng of RNA from each sample was treated with DNAase I for 30 min at 37 °C (Thermo Scientific, Waltham, MA, USA) and used to synthesize cDNA using Maxima Reverse Transcriptase (Thermo Scientific, USA). Real-time quantitative PCR was performed on a Light Cycler 480 II (Roche) using the TB Green Premix Ex Taq II (Tli RNase H Plus) (TaKaRa, Kusatsu, Japan). For this purpose, 2.5 µL of the cDNA mixture was added to each reaction containing 0.4 µM of the corresponding oligonucleotides (Table 2) and 7.5 µL of the RT-PCR MIX in a final volume of 10 µL. A standard curve was constructedwith serial dilutions of the cDNA sample (2 × 10^−1^, 1 × 10^−1^, 2 × 10^−2^, 1 × 10^−2^, 2 × 10^−3^, and 1 × 10^−3^). As an internal control, GAPDHq_F and GAPDH_R oligonucleotides were used to determine GAPDH mRNA levels.

### 2.9. ACE-2/Spike Antibody Inhibition Assay

To determine the ability of the vaccination to produce neutralizing antibodies, the presence of neutralizing antibodies was tested in serum collected from both donors after vaccination using a hACE-2/spike antibody inhibition ELISA-based assay. Microtiter 96-well plates were coated with a chimeric monoclonal anti-Foldon antibody [32] overnight at 8 ng/µL in PBS (10.1 mM Na_2_HPO_4_, 1.5 mM KH_2_PO_4_, 2.7 mM KCl, 137 mM NaCl). The next day, plates were blocked with PBS supplemented with BSA fraction V (Sigma) at 1% (PBS-BSA 1%). Then, purified HexaPro-derived construct containing D614G substitution was captured by incubation at 1 ng/µL in PBS-BSA 1% over the course of45 min RT [33]. Following protein incubation, plates were washed with PBS, and successive incubations at RT of sera dilutions and hACE-2 monomeric-StreTag receptor (20 ng/µL) complexed with StrepTactin-HRP (1:5000) were performed. Sera incubation was prolonged for 45 min, and after 15 min of incubation of receptor-StrepTactin HRP complexes, receptor binding to captured spike was revealed with the OPD substrate (Sigma Aldrich) and measured in a spectrophotometer reading OD at 493–620 nm. The assay background was determined in parallel using a spike protein locked in the closed conformation, which is unable to bind the hACE-2 receptor. For this purpose, a purified HexaPro-derived construct, including a double cysteine substitution, S383C D985C, was used. A pool of sera from individuals negative for anti-SARS-CoV-2 antibodies and hACE-2 monomeric untagged receptor were used as negative and positive controls, respectively. After subtraction of the background, the percentage of neutralization was calculated as [1 − (OD_495–620_ test serum/OD_495–620_ negative control)] × 100%. In all experiments, incubation of hAE-2 untagged receptor at 200 ng/µL achieved a neutralization rate higher than 85%.

### 2.10. Statistical Analysis

The cytotoxicity results were analyzed using GraphPad 9.0 software, and the results areexpressed as a mean and standard deviation. SARS-CoV-2 mRNA expression and cytotoxicity levels were compared using tailed *t*-student tests. The asterisk (*) indicates statistically significant differences with *p*-value ≤ 0.05.

## 3. Results

### 3.1. SARS-CoV-2 Antigen-Specific T-Cell Production and Reactivity

The characteristics of the donors and apheresis are summarized in Table 3. At the time of the first lymphapheresis, the leukocyte count was 3.6 and the lymphocyte countwas0.9 (×10^6^/mL) for donor 1, and for donor 2, the leukocyte countwas 5.9 and the lymphocyte count was 1.7 (×10^6^/mL). In the second set of blood samples, the white blood cell count was 4.5 and the lymphocyte count was 1.3 (×10^6^/ mL) for donor 1, and for donor 2, the leukocyte count was 6.5 and the lymphocyte count was 2.0 (×10^6^/mL) (Table 3).

Donor 1 showed greater in vitro reactivity against the M protein before receiving the RNA vaccine. After vaccination, in addition to showing reactivity against the M protein (0.09%), donor 1also showed reactivity against the S (0.06%) protein (Figure 1B). Based on theresults obtained using the M-protein peptide mix during screening, 1 × 10^9^ leukocytes were used toobtain M-protein SARS-CoV-2-specific T-cells. After purification, the material from donor 1 before vaccination contained 1.35 × 10^6^IFNγ+ CD3+ T-cells with 7.42% IFNγ+ CD3+ T-cell enrichment (Figure 2A). In addition, using the blood obtained after vaccination, the number of IFNγ+ CD3+ T-cells in the final material was enrichedby 70.47% (Figure 2A), obtaining 2.4 × 10^6^ IFNγ+ CD3+ purified T-cells. Reactivity against SARS-CoV-2 peptides did not change significantly.

Donor 2 showed reactivity against the three tested proteins (M, N, and S). We observed that, after receiving the vaccine, donor 2 showed increased reactivity against all three proteins: 0.03%, 0.03%, and 0.06% against M, N, and S, respectively (Figure 1B). Thus, a pool of peptide mixes covering all three M, N, and S proteins was used for enrichment for donor 2. After purification, the collected material before vaccination contained 1.57 × 10^6^IFNγ+ CD3+ T-cells, obtaining 30.19% IFNγ+ CD3+ T-cell enrichment. Using the blood sample after vaccination, we obtained an enrichment of 42.59% IFNγ+ CD3+ T-cells (Figure 2B), obtaining 2.46 × 10^6^IFNγ+ CD3+ purified T-cells. Antigen-specific T-cells exhibited higher reactivity against M (0.12%), N (0.06%), and S (0.24%) proteins.

The populations of CD4 and CD8 T-cells were analyzed in peripheral blood lymphocytes and SARS-CoV-2 antigen-specific T-cells from apheresis before and after vaccination. The proportion of CD4 T lymphocytes was higher in samples from both donors. This percentage increased approximately 20% after the production of SARS-CoV-2 antigen-specific T-cells in both samples collected before vaccination, with a similar decrease in the proportion of CD8 T-cells (Table 4). However, in samples collected after vaccination, we observed a significant increase in the proportion of CD8 T-cells (Table 4).

### 3.2. Transfection Efficiency and Expression of M Protein

Since in vitro reactivity was predominant against the M protein, we generated an HEK 293T cell model transiently expressing the SARS-CoV-2 membrane (M) protein to test M-protein-mediated specific cytotoxic activity. As a transfection control, HEK 293T cells were transiently transfected with a GFP protein reporter plasmid in parallel. Our results indicate that 73.33% ± 17.48 of cells expressed the GFP protein, indicating optimal performance of the transfection procedure. Figure 3A shows a representative example of the transfected cells. We also analyzed the expression of the SARS-CoV-2 M protein in the transfected cells by RT-PCR. As shown in Figure 3B, the expression of the M protein in the cells transfected with the pcDNATM3.1-M-gene construct was similar (0.08 ± 0.05) to that obtained for the constitutive gene GAPDH (value = 1), while cells transfected with the empty plasmid had no expression of the M protein.

### 3.3. M-Protein-Mediated Specific Cytotoxicity Activity and Antibody Titers

The T-cell-mediated cytotoxicity assay was performed using stimulated and unstimulated T-cells from both donors at pre- and post-vaccination time points, and the results are shown in Figure 3C. The cells isolated from both donors had similar cytotoxicity patterns. Unstimulated T-cells showed no or low cytotoxic activity at both pre- and post-vaccination time points. However, SARS-CoV-2 peptide-stimulated T-cells collected from COVID-19convalescent patients at the pre-vaccination time point had significantly higher cytotoxic activity (lysing the M-protein-expressing HEK 293T cells) than that of theunstimulated control (donor 1: 8.55%, *p*-value 1.44 × 10^−5^; donor 2: 33.38%, *p*-value 3.13 × 10^−6^). In addition, the observed cytotoxic effect of the SARS-CoV-2 peptide-stimulated T-cells was significantly increased after vaccination compared with that observedat pre-vaccination (donor 1: 32.71%, *p*-value 4.10 × 10^−4^; donor 2: 33.38%, *p*-value 8.70 × 10^−4^).

Post-vaccination serum samples from both COVID-19 convalescence patients were analyzed in order to determine whether a neutralizing antibody response was elicited after vaccination, which is indicative of the activation of a specific SARS-CoV-2 immune response. Both patients exhibited Ab titers, 120 and 190 for donors1 and 2, respectively, concordant with those obtained after vaccination by other groups [34], which confirmed an optimal response of both donors to vaccination.

## 4. Discussion

The T-cell immune response has been shown to be a key factor for the control of both virus clearance and the severity of COVID-19, in addition to the humoral immune response [13,35,36]. Thus, T-cell adoptive transfer could be an alternative therapy to treat COVID-19 patients. To achieve this, it is crucial to demonstrate the feasibility of preparing clinical-grade SARS-CoV-2-specific T-cells from convalescent donors and the ability of these cells to neutralize the virus in vitro.

In this study, we were able to quickly and efficiently isolate SARS-CoV-2-specific T lymphocytes from COVID-19 convalescent donors through stimulation with specific SARS-CoV-2 peptides followed by automated isolation using the CliniMacs Prodigy medical device. This method is simple, and it can be performed overnight so that the material used in treatment canbe available within 24 h. Consistent with other studies, the proliferation of SARS-CoV-2-specific CD4^+^ T lymphocytes was obtained after ex vivo activation in the CliniMACS Prodigy 3 [37,38]. Our results demonstrate that the generated cells using the CliniMACS Prodigy system are highly specific against the SARS-CoV-2 virus, similar to results previously foundfor other viruses [39]. The stimulation and subsequent selection system with IFN-γ has previously been validated for obtaining specific antigen cells [40,41].

Although functional SARS-CoV-2-specific T lymphocytes were successfully isolated from COVID-19 convalescent donors before and after receiving the SARS-CoV-2 mRNA Pfizer-BioNTech vaccine, significantly higher levels of SARS-CoV-2-specific T-cells were obtained after vaccination, suggesting that this donor profile might be more suitable for this purpose. In addition, our results suggest that the SARS-CoV-2 M protein induces a higher specific T-cell response. In fact, other vaccines with multiple targets are currently under development with the aim of inducing a broad T-cell response [15,42]. As such, UB-612, a protein-based vaccine that includes multiple epitopes of the matrix (M), S2, and nucleocapsid (N) proteins of SARS-CoV-2, is currently being tested in a phase II/III clinical trial (NCT04683224).

To further characterize the ability of M-protein-stimulated T-cells to lyse target cells, we implemented a transfection strategy to express the SARS-CoV-2 M protein in HEK 293T cells mimicking SARS-CoV-2 infection. Using this approach, we demonstrated that the isolated SARS-CoV-2-specific T-cells had the ability to selectively recognize and lyse cells expressing the SARS-CoV-2 M protein. Interestingly, we also observed a bystander stimulation of activated T-cells of unrelated specificity following vaccination, since there was an increase in the M-protein-specific T-cells upon vaccination, even thoughthe vaccine is based on the SARS-CoV-2 spike protein. This effect has previously been observed using other vaccines, such as the tetanus toxoid vaccine [43].

While the administration of mRNA vaccines has effectively reduced COVID-19-related severe disease, hospital admissions, and mortality in healthy adults [44,45,46], vaccine efficacy has not been as high in other target populations. In fact, different studies have reported lower rates of immune responses after vaccination in patients with hematologic pathologies (chronic lymphatic leukemia, multiple myeloma, lymphoma, and myeloproliferative malignancies) [47,48,49,50], as well as in solid organ transplant recipients [51,52,53,54]. In this scenario, it may be interesting to have new alternative therapies available that focus on increasing the individual immune response against the SARS-CoV-2 infection.

Thus, patients with more severe forms of COVID-19, including respiratory failure, requiring hospitalization and oxygen therapy, may be ideal candidates for adoptive T-cell therapy, since they mostly develop severe lymphopenia (lymphocyte count <0.8 × 10^9^/L) [22,55,56,57]. A possible limitation of this study is that further studies are needed to assess the clinical utility of the functional isolated SARS-CoV-2-specific T-cells in patients since using adoptive T-cell therapy may be limited to human leukocyte antigen (HLA)-mismatched virus-specific T-cells [58,59]. However, the clinically effective dose used is around 5 × 10^3^ virus-specific T-cells/kg, which is considerably lower than the clinical threshold for graft-versus-host disease in transplant recipients. Although it is recommended that SARS-CoV-2-specific T-cells from the donor must share some HLAs with the recipient for proper presentation of the viral antigen to donor T-cell antigen receptors, previous studies have revealed that the use of virus-specific T-cells from donors sharing only one HLA was effective to treat severe viral infections after hematopoietic stem cell transplantation [60].

Previous studies from patients recovered from SARS-CoV-2 consider the antibody response where SARS-CoV-2-specific IgM and IgA lasted less than 6 months and specific IgG titers markedly declined after 1 year [61]. Despite the lack of specific memory B cells, SARS-CoV-2-specific memory T-cells persisted in SARS-recovered patients for up to 6 years post-infection. Therefore, a strong virus-specific T-cell response is required to protect against fatal SARS-CoV-2 infection, and, for this reason, virus-specific T-lymphocyte infusion may be considered as a potential treatment for these patients [15,62].

## 5. Conclusions

We demonstrated that SARS-CoV-2-specific T-cells can quickly and efficiently be stimulated from the blood of convalescent donors using SARS-CoV-2-specific peptides followed by automated isolation. We also demonstrated that vaccinated convalescent donors have a higher percentage of SARS-CoV-2-specific T-cells and may be more suitable as donors. Although further studies are needed to assess the clinical utility of the functional isolated SARS-CoV-2-specific T-cells, previous studies using the same stimulation and isolation system applied to other pathologies support this approach.

## Figures and Tables

**Figure 1 biomedicines-10-00630-f001:**
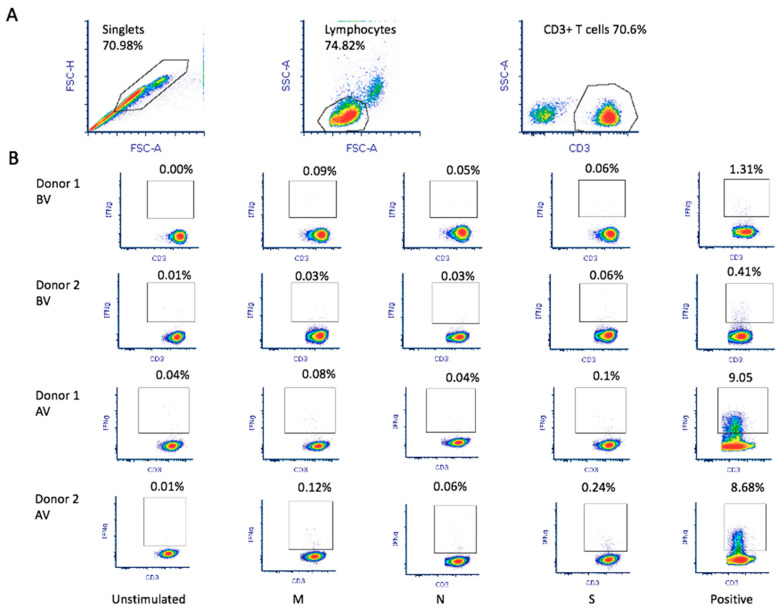
SARS-CoV-2 antigen-specific T-cell production and reactivity werefirst tested using 10 mL of whole peripheral blood collected from each donor in EDTA tubes. (**A**) Flowcytometry analysis was based on doublet exclusion before selection of the population of interest (lymphocytes) and CD3+ T-cells. A minimum of 100,000 events within the population of interest were analyzed. (**B**) The percentage of CD3+ IFN-γ+ cells was quantified upon stimulation with the different SARS-CoV-2 peptides (M, N, and S). For each experiment, an unstimulated sample was used as negative control and a non-specific stimulation was used as positive control. Triplicate experiments were performed for both donors before vaccination (BV) and after vaccination (AV).

**Figure 2 biomedicines-10-00630-f002:**
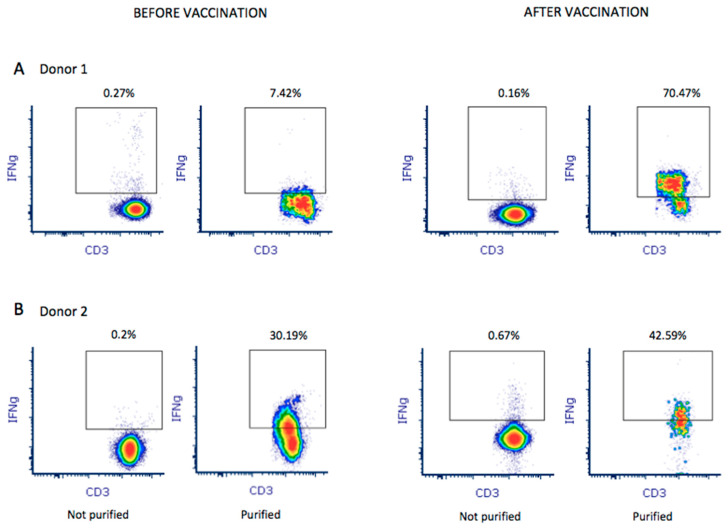
A total of 1 × 10^9^ leukocytes were used to purify and enrich M-protein SARS-CoV-2-specific T-cells from both donors before and after vaccination. Enrichment of IFNγ+ CD3+ T-cells in the final product was analyzed by flow cytometry for donor 1 (**A**) and donor 2 (**B**). Percentage of enrichment was quantified by establishing the quality control before purification as baseline (not purified).

**Figure 3 biomedicines-10-00630-f003:**
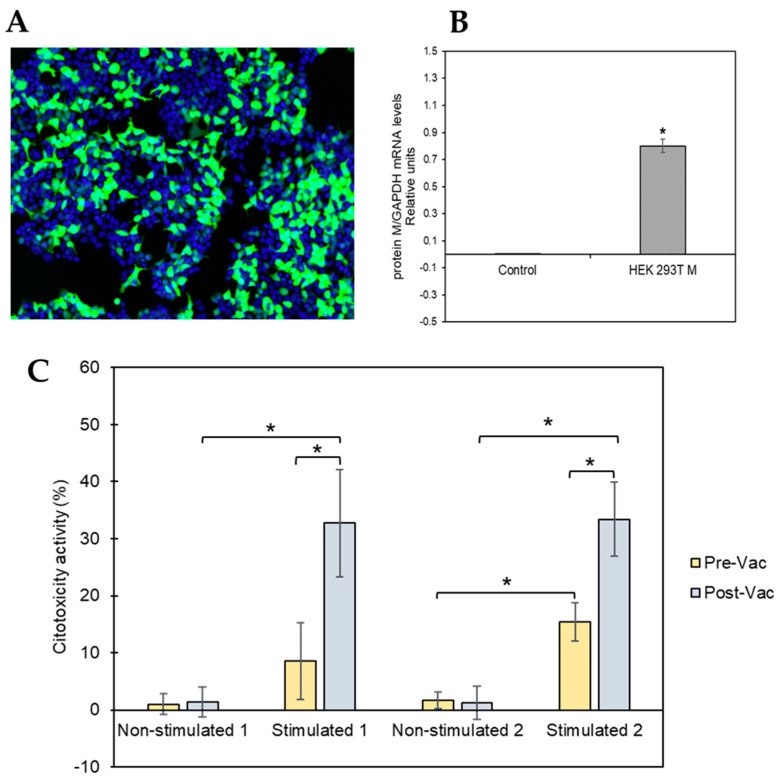
HEK 293Tcell model to test M-protein-mediated specific cytotoxicity activity. (**A**) Representative image of the transfected cells. Nuclei are shown in blue (Hoechst 33342), while transfected cells are visible in green (GFP). (**B**) Expression levels of SARS-CoV-2 mRNA for M protein were measured 48 h post-transfection of HEK 293T cells using RT-qPCR. The obtained results were normalized to the mRNA levels of the constitutive gene GAPDH. The asterisk (*) indicates that levels of SARS-CoV-2 mRNA for M protein were significantly higher (*p*-value ≤ 0.05, tailed t-student test) in HEK 293T cells transfected with the plasmid pcDNATM3.1/myc-His containing the M protein coding gene than the levels obtained after transfecting the HEK 293T cells with the control plasmid (pcDNATM3.1). (**C**) Percentage of cytotoxic activity of the stimulated and non-stimulated T-cells against M protein of SARS-CoV-2 expressed in HEK 293Tcells in the membrane. T-cells isolated from two convalescent donors before and after vaccination were isolated for the cytotoxicity assay. All the experiments were performed at least in triplicate at both time points (before and after vaccination).

**Table 1 biomedicines-10-00630-t001:** List of fluorochrome-labeled antibodies used in flow cytometry studies. FITC, fluorescein isothiocyanate; PE, phycoerythrin; PE/Cy7, tandem comprising phycoerythrin and cyanine 7; APC, allophycocyanin; APC/Cy7, tandem comprising allophycocyanin and cyanine 7.

Antigen	Clone	Fluorochrome	Source	Catalog Number
CD3	UCHT1	PE/Cy7	Biolegend	351304
INF-γ	45-15	PE	Miltenyi Biotec	130-113-493
CD4	OKT4	APC/Cy7	BD Pharmingen	317418
CD8	SK1	PerCP/Cy5.5	Biolegend	344710
DAPI			Sigma-Aldrich (Merck)	D9542

**Table 2 biomedicines-10-00630-t002:** List of oligonucleotides used in the study.

Oligonucleotides	Sequence (5′→3′)
ProtM_XhoI_F	CCGCTCGAGCGGCCACCATGGCAGATTCCAACGGTAC
ProtM_KpnI_R	CGGGGTACCCCGTTACTGTACAAGCAAAGCAA
ProtMseq_F	GTAGGCGTGTACGGTGGGAG
ProtMseq_R	CAGTCGAGGCTGATCAGCGG
ProtMq_F	GCCACTCCATGGCACTATT
ProtMq_F	GTATTGCTGGACACCATCTAGG
GAPDHq_F	GGTGTGAACCATGAGAAGTATGA
GAPDHq_R	GAGTCCTTCCACGATACCAAAG

**Table 3 biomedicines-10-00630-t003:** Donor apheresis parameters of SARS-CoV-2 convalescent donors before and after receiving the Pfizer-BioNTech vaccine. TVP: total volume processed; ACD: acid-citrate-dextrose; TNC: total nucleated cells; CE: collection efficiency.

	Before Vaccination		After Vaccination	
	Donor 1	Donor 2	Donor 1	Donor 2
Date of apheresis	28 May 2020	14 December 2020	26 March 2021	06 April 2021
TVP	6819 mL	2370 mL	1981 mL	2441 mL
ACD anticoagulant	684 mL	230 mL	180 mL	203 mL
Product volume	138 mL	36 mL	40 mL	45 mL
TNC in product	7.75 × 10^9^	2.6 × 10^9^	2.88 × 10^9^	2.49 × 10^9^
Hematocrit	2.9%	3%	2.7%	2.3%
CE	43%	7%	13%	7%

**Table 4 biomedicines-10-00630-t004:** CD4/CD8 T-cells. Percentage of CD4 and CD8 T-cell populations analyzed in peripheral blood lymphocytes and SARS-CoV-2 antigen-specific T-cells from aphaeresis before vaccination and after vaccination.

	Peripheral Blood T-Cells				SARS-CoV-2 Antigen Specific T-Cells			
	Before Vaccination		After Vaccination		Before Vaccination		After Vaccination	
	CD4	CD8	CD4	CD8	CD4	CD8	CD4	CD8
Donor 1	76.27%	21.47%	66.5%	27.44%	94.27%	4.8%	82.9%	9.35%
Donor 2	49.98%	45.06%	40.5%	55.5%	69.76%	27.18%	45.51%	49.2%

## Data Availability

The data that support the findings of this study are available from the corresponding author upon reasonable request.

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
