# Peer review of "Isolation of Functional SARS-CoV-2 Antigen-Specific T-Cells with Specific Viral Cytotoxic Activity for Adoptive Therapy of COVID-19"

_biomedicines, 2022, doi:10.3390/biomedicines10030630_

Round 1

Reviewer 1 Report

An excellent small study on a topic that is rarely undertaken in COVID-19, hence I consider the manuscript a valuable achievement for the authors, worth being published in Biomedicines.
I have no comments on the methodology or results - everything is clear and logical order here.
Discussion - limitations should be described.
References need to be completed; here are two examples: doi: 10.3390/pathogens9060493.; doi: 10.1055/a-1535-8807.

Author Response

Dear Dr. Yi,

Thank you for your comments and for those of the reviewers regarding our recently submitted manuscript. We appreciate the time and effort that have gone into reviewing our manuscript and for the constructive comments of the reviewers.

We have modified the paper, taking into account the reviewers’ comments. A point-by-point response is included below.

We hope that the manuscript is now acceptable for publication.

We look forward to your response.

Sincerely,

Pilar Pérez-Romero

Response to reviewer 1:

An excellent small study on a topic that is rarely undertaken in COVID-19, hence I consider the manuscript a valuable achievement for the authors, worth being published in Biomedicines.

I have no comments on the methodology or results - everything is clear and logical order here.

Discussion - limitations should be described.

References need to be completed; here are two examples: doi: 10.3390/pathogens9060493.; doi: 10.1055/a-1535-8807.

Response: We appreciate the reviewer’s positive comments and suggestions. Following the reviewer’s suggestion, we have included new references in the introduction and a paragraph in the discussion highlighting the limitations of the study.

Reviewer 2 Report

The current manuscript tried to collect COVID-19 specific T cells and detect the cytotoxic activity in vitro. The idea is novel, but the data had defect and was limited, it must be improved before accept.

  1. How did you do stimulation in vitro? Based on your methods, it is difficult to see IFN-R, you have to add some costimulatory factors. Did you do repeat? if only one time experiment, it is not enough to say anything.
  2. Figure 3B, how did you compare these two groups? I can not understand.
  3. there were not enough information to understand the meaning of current Figures, and it is important to show the "n=" in each figure. 

Author Response

Dear Dr. Yi,

Thank you for your comments and for those of the reviewers regarding our recently submitted manuscript. We appreciate the time and effort that have gone into reviewing our manuscript and for the constructive comments of the reviewers.

We have modified the paper, taking into account the reviewers’ comments. A point-by-point response is included below.

We hope that the manuscript is now acceptable for publication.

We look forward to your response.

Sincerely,

Pilar Pérez-Romero

Response to reviewer 2:

The current manuscript tried to collect COVID-19 specific T cells and detect the cytotoxic activity in vitro. The idea is novel, but the data had defect and was limited, it must be improved before accept.

  1. How did you do stimulation in vitro? Based on your methods, it is difficult to see IFN-R, you have to add some costimulatory factors. Did you do repeat? if only one time experiment, it is not enough to say anything.

Response: We appreciate the reviewer’s comments and suggestions. We have included additional information in the Methods section regarding the in vitro stimulation of the T-cells. To respond the reviewer´s concern, at least triplicate experiments were performed in both donors, at two different time points, before and after vaccination. To clarify this issue, this information was also included in the figure legends.

  1. Figure 3B, how did you compare these two groups? I can not understand.

Response: In response to the reviewer´s question, Figure 3B represents the quantitation of the expression of SARS CoV-2 mRNA for M protein, using RT-qPCR in HEK 293T cells transfected with the plasmid pcDNATM3.1/myc-His containing the protein M gene, compared with HEK 293T cells transfected with the control plasmid (pcDNATM3.1). SARS CoV-2 mRNA expression levels were compared using tailed t-student tests. The asterisk (*) indicates statistically significant differences with p-value ≤ 0.05. A paragraph including this information was added in the legend of Figure 3 and in the methods section.

  1. there were not enough information to understand the meaning of current Figures, and it is important to show the "n=" in each figure. 

Response: In response to the reviewer´s suggestion and in order to facilitate the understanding of the Figures, additional information was included in the figure legends.

Round 2

Reviewer 2 Report

Authors already addressed my comments.